

# On the benefits of clustering approaches in digital soil mapping: an application example concerning soil texture regionalization

István Dunkl[1,2] and Mareike Ließ[2]

[1]Max Planck Institute for Meteorology, Hamburg, Germany
[2]Department Soil System Science, Helmholtz Centre for Environmental Research – UFZ, Halle (Saale), Germany

**Correspondence:** István Dunkl (istvan.dunkl@mpimet.mpg.de)

**Abstract.** High resolution soil maps are urgently needed by land managers and researchers for a variety of applications. Digital Soil Mapping (DSM) allows to regionalize soil properties by relating them to environmental covariates with the help of an empirical model. In this study, a legacy soil data set was used to train a machine learning algorithm in order to predict the particle size distribution within the catchment of the Bode river in Saxony-Anhalt (Germany). The ensemble learning method random forest was used to predict soil texture based on environmental covariates originating from a digital elevation model, land cover data and geologic maps. We studied the usefulness of clustering applications in addressing various aspects of the DSM procedure. To investigate the role of the imbalanced data problem in the learning process, the environmental variables were used to cluster the landscape of the study area. Different sampling strategies were used to create balanced training data and were evaluated on their ability to improve model performance. Clustering applications were also involved in feature selection and stratified cross-validation. Overall, clustering applications appear to be a versatile tool to be employed at various steps of the DSM procedure. Beyond their successful application, further application fields in DSM were identified. One of them is to find adequate means to include expert knowledge.

## 1 Introduction

In order to sustain soil resources, land managers and researchers are in need of information on the continuous landscape-scale distribution of soil properties. One of the important soil properties which governs most physical, chemical, and biological soil processes is soil texture. Soil texture maps can be used for the assessment of erosion risk, water deficit, or pesticide and nutrient storage and percolation.

Conventional soil maps are usually created by a qualitative analysis of the landscape based on a conceptual model which subdivides the area into spatially assigned units with all soil properties set to uniform values within the units. The categories of these units do not necessarily represent soil systematic units and do not allow for the representation of small-scale, continuous variability. Overall, these soil maps were never meant to be used as input to landscape-scale process models that strive to simulate gas, matter and water flows. From this demand and an advance in information technology, the domain of Digital Soil Mapping (DSM) has quickly advanced (Grunwald et al., 2011).



DSM strives to capture and quantify the influence of the soil forming factors, which are represented by continuous gridded
geo-information from remote sensing and other sources (Scull et al., 2003). Laboratory and field observations are coupled
with spatial environmental covariates covering the study area and are used to build an empirical model to predict the surveyed
response variable based on the quantitative relationship between soil properties and environmental covariates (McBratney
et al., 2003; Grunwald, 2009; Minasny and McBratney, 2016). The key technological advantages that allowed DSM are the
increase in computational power which facilitates model development, and the widespread availability of satellite systems
(Rossiter, 2018). The latter are used for accurate georeferencing and as platforms for a variety of sensors which provide
spatially continuous measurements which can be used as environmental covariates.

The algorithms used for DSM applications are of different degrees of complexity ranging from regression analysis (Gobin
et al., 2001; Park and Vlek, 2002; de Carvalho Junior et al., 2014) to artificial neural networks (Park and Vlek, 2002; Zhao
et al., 2009). Most of these studies used continuous predictors based on a digital elevation model (DEM), but certain appli-
cations also included categorical predictors, such as information based on geologic maps (Adhikari et al., 2013; Vaysse and
Lagacherie, 2017). The machine learning algorithm most frequently used in DSM approaches is random forest (RF) ensemble
learning method (e.g. Blanco et al. (2018); Padarian et al. (2019); Møller et al. (2019)). A key characteristic of RF is its adap-
tive nature which allows it to explore complex, nonlinear, and high-dimensional relationships, without a prior understanding
of the problem to be solved (Evans et al., 2011). Compared to parametric methods, RF is not prone to overfitting, even in the
presence of some irrelevant parameters and outliers (Heung et al., 2014). Nevertheless, many RF applications and most other
modelling applications use feature selection preceding the model building procedure. Feature selection reduces the noise intro-
duced through uninformative predictors. This can be achieved though filter methods, which investigate the predictor-response
relationship of each predictor individually without considering the model algorithm or alternatively by using wrapper functions
that evaluate the performance of the model using a variety of predictor subsets. Feature selection can, however, also include
the omission of strong predictors, as they might dominate the model output and cause the emergence of artifacts.

The essential foundation of creating soil maps is the availability of a soil dataset of sufficient size and adequate distribu-
tion, but the soil surveys providing this data are associated with high cost and labor (Grunwald et al., 2011). To forego this
effort, DSM is using legacy soil data whenever available. However, sampling in traditional soil surveys usually did not follow
statistical sampling theory which can lead to a bias in the data and the models derived from it (Carré et al., 2007). Because
soil forming factors operate on different scales, it is important that the spatial distribution of the data is suitable to capture
the large- and small-scale variation of soil. Moreover, a bias could be added to the prediction if samples from certain parts of
the landscape are over- or under-represented in the data. This would lead to an imbalanced learning problem and compromise
the predictive performance of the models (He and Garcia, 2008). In order to construct a model that can effectively predict
throughout the landscape, it is important to have a statistically representative sample of training and validation data that allows
for the generalisation from the data to the spatial landscape context (Ließ, 2020). The most common approaches in dealing
with this issue involve (a) creating a more balanced training set by sampling from the entirety of observations, and (b) cost-
sensitive learning frameworks, in which the learning algorithm penalizes the prediction error of underrepresented samples (He
and Garcia, 2008). Many DSM applications tackle the problem of data imbalance with the subsampling approach (Moran and





Bui, 2002; Subburayalu and Slater, 2013; Heung et al., 2016; Sharififar et al., 2019). This can be achieved by stratifying the
study area into homogeneous sections, and drawing a certain number of samples from each of these strata.

Another hurdle of modeling applications lies in training and tuning. Model building and performance evaluation can be
sensitive to the selection of training and testing samples. Although resampling techniques like cross-validation (CV) increase
model robustness, the outcome can still be compromised by an uneven distribution of classes between the data subsets.

Many of these challenges in the DSM procedure are related with identifying structures and similarities in the data. Therefore,
here we want to investigate the usefulness of data clustering applications in tackling some of the above mentioned challenges
in DSM. Specifically, we want to examine the benefits of using clustering applications for feature selection, to address the
imbalanced learning problem, and resampling to build robust models. This will be done on the basis of training an RF model
to predict soil texture within the catchment of the Bode river in Saxony-Anhalt, Germany. The model is trained and validated
using a soil legacy data set containing soil survey data. Environmental covariates related to soil forming factors are obtained
and used as predictors.

## 2 Material and methods

### 2.1 Study area and data

#### 2.1.1 Study Area

The study area of approximately 3,3000 km$^2$ is part of the TERENO network for environmental observations (Zacharias et al.,
2011) and covers the water catchment of the river Bode in central Germany (Fig. 1). It corresponds to three federal German
states: Saxony-Anhalt, Lower Saxony and Thuringia. The elevation ranges between 1 and 1141 m a.s.l. with the Harz Mountains
in the southwest, the north-eastern Harz foreland and the Magdeburg Börde of the North German Plain covering the rest of the
area. The climate is subarctic to humid continental (Peel, Finlayson and McMahon, 2007), with the mean annual precipitation
ranging from 433 to 1771 mm (Deutscher Wetterdienst, 2019). The geologic material in the area consists mostly of Triassic
limestone and Carboniferous shale and granite (BGR, 2007). Dominating soils according to the German soil classification
(Finnern and Kühn, 1994) are Braunerde, Parabraunerde, Gley and Pararendzina (BGR, 2012).

#### 2.1.2 Soil legacy data

The soil samples used for model training and validation are from a legacy data set provided by the regional geological survey
of the German federal state Saxony-Anhalt - Landesamt für Geologie und Bergwesen (LAGB, 2018). The data was recorded by
various soil surveyors between 1963 and 2006 and consists of soil profile data from 574 sites. For every site, a soil diagnostic
survey was conducted. Soil horizon boundaries were recorded according to either the TGL (TGL, 1985), or the KA4 (Finnern
and Kühn, 1994) soil systematic system. For every soil horizon, the particle size distribution was measured in the laboratory
using DIN ISO 11277:2002-08. The fractions of three particle sizes were measured according to the German soil separates
(sand [2 mm to 0.063 mm], silt [0.063 mm to 0.002 mm], and clay [< 0.002 mm]). Sand, silt and clay contents were extracted

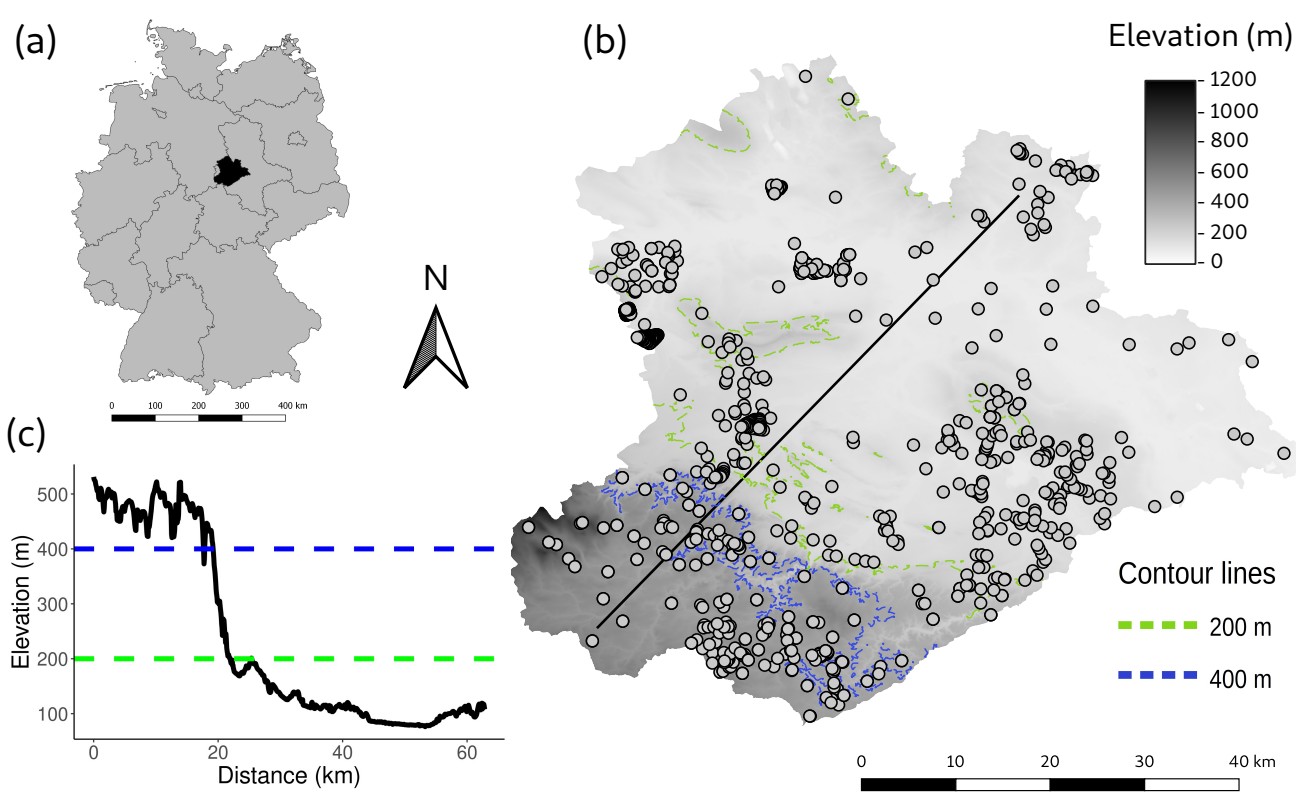

**Figure 1.** Study area (a) location in Germany, (b) location of the survey sites, (c) cross-sectional elevation profile (black line in map).

from the horizon data at two discrete soil depths (10 and 70 cm) and used as the target variables of the models. The two depths were chosen to investigate whether different soil forming factors dominated soil-landscape development in the topsoil and subsoil, respectively. Because the maximum depth of the surveyed soil profiles is not uniform, a depth of 70 cm was chosen as a trade-off between maximum soil depth (closeness to parent material), while not compromising the sample size. One sample was removed from the profiles as an outlier. The sample is located in a Quaternary sand dune of less than 2 km² (BGR, 2007) near the town of Blankenburg, and has a sand content of 96 %. The sample was removed because one sample alone would not be sufficient for model training and validation. The soil texture of the soil legacy dataset used as model input is shown in Fig. 2 a and b. A cluster analysis targeting three equally sized subgroups was applied to differentiate clayey samples from silty and sandy samples, please refer to the section Cluster analysis for details. Figure 2 c and d show the spatial distribution of these three clusters within the study area.





(a)

(b)

(c)

(d)

**Figure 2.** Soil legacy dataset used for model development. Particle size distribution and cluster affiliation of the soil data at (a) 10 and (b) 70 cm depth, respectively. (c) and (d) show the spatial distribution of the three clusters at 10 and 70 cm depth (Geographic coordinate system: UTM zone 32N)

### 2.1.3 Model predictors


Spatially continuous geodata of the study area corresponding to the soil forming factors parent material, topography and land cover were gathered. They comprise geologic maps of 1:200,000 (GUEK200) and 1:1,000,000 (GUEK1000) map scale (BGR,





2007, 2006), a DEM of 10 m resolution (BKG, 2012), and CORINE Land Cover data from 1990, 2000, and 2012 (Büttner et al., 2004). The local river network was generated from the OpenStreetMap data set by querying rivers and streams with the

Overpass API service (OpenStreetMap contributors, 2018). Some of the geodata was used without further modification like the land cover data and the elevation from the DEM. Additionally, further variables were derived from these data and have been resampled to the 10 m resolution of the DEM.

In the digital vector information underlying the geologic maps, a variety of attributes is contained including age, material, and origin. The layer 'petrography' was used from both geologic maps, and the layer 'genesis' was used from the 1:200,000

map. As the information contained in the petrography layer is descriptive, it was categorized into binary information on the occurrence of particle size classes in addition to its inclusion as unmodified layer. Three new predictors (Sandbin, Siltbin and Claybin) were created for every landscape unit based on the occurrence of the words sand or sandstone, silt or siltstone, and clay or claystone, respectively.

Several topographic predictors were derived from the DEM, since relief is often considered as the main driver of soil

formation (McBratney et al., 2003; Scull et al., 2003; Behrens et al., 2010). Topographic predictors were calculated with the SAGA GIS software Version 6.4.0 (Conrad et al., 2015). The used topographic predictors were selected according to their appearance in similar digital soil mapping applications (Bulmer et al., 2016; Vaysse and Lagacherie, 2017; Blanco et al., 2018; Kalambukattu et al., 2018; Zhou et al., 2019) Table 1. Sink removal by Wang and Liu (2006) was applied prior to the calculation of the hydrological terrain parameters (minimal slope = 0.01). For the calculation of the vertical distance to channel network,

the layer of waterways acquired from OpenStreetMap was used. Indices for terrain convexity and terrain surface texture were calculated by using a flat area threshold of 0.08 in order to minimize the impact of inaccuracies and insignificantly small depressions and mounds (Conrad et al., 2015).

Since soil-forming factors can take effect on different spatial scales, it is advised to take multi-scale approaches into account (Behrens et al., 2010). Accordingly, convergence index (Köthe and Lehmeier, 1996), terrain ruggedness index (Riley, 1999),

convexity and terrain surface texture were calculated with a search radius of 10, 50, 100 and 200 m, in order to express local to regional landscape attributes. The annulus based topographic position index (Guisan et al., 1999) was calculated on two scales, one ranging from 0 to 100 m and from 100 to 200 m.

## 2.2 Modeling procedure

### 2.2.1 Random forest

RF models are based on regression trees (RTs), which use selected values of predictor variables to repeatedly split the data in a way that maximizes the homogeneity of the subsets regarding the response variable (Kuhn and Johnson, 2013). Besides the benefit of their good interpretability, RTs have several shortcomings in terms of model performance (Evans, 2009). One reason for the limited performance lies in the recursive splitting of the data. This splitting assumes the homogeneity within rectangular regions in the predictor space and returns discrete outcome values for these regions. Another issue of regression trees is their





**Table 1.** Topographic predictors derived from the digital elevation model. Indices that have been calculated with varying parameters are denoted by multiscale.

| Domain | Predictor | Reference |
|---|---|---|
| Morphometry | Slope | Zevenbergen and Thorne (1987) |
| | Convergence index (multiscale) | Köthe and Lehmeier (1996) |
| | Mass balance index | Friedrich (1996) |
| | Slope height | Böhner and Selige (2006) |
| | Normalized height | Böhner and Selige (2006) |
| | Standardized height | Böhner and Selige (2006) |
| | Valley depth | Böhner and Selige (2006) |
| | Mid-slope position | Böhner and Selige (2006) |
| | Terrain ruggedness index (multiscale) | Riley (1999) |
| | Convexity (multiscale) | Conrad et al. (2015) |
| | Terrain surface texture (multiscale) | Conrad et al. (2015) |
| | Multi-Scale Topographic position index (multiscale) | Guisan et al. (1999) |
| Lighting | Positive openness | Yokoyama et al. (2002) |
| | Negative openness | Yokoyama et al. (2002) |
| | Diffuse insolation | Böhner and Antonić (2009) |
| | Direct insolation | Böhner and Antonić (2009) |
| Hydrology | Terrain classification index for lowlands | Bock et al. (2007) |
| | LS-Factor | Böhner and Selige (2006) |
| | Stream power index | Moore et al. (1991) |
| | Topographic wetness index | Beven and Kirkby (1979) |
| | Upslope contributing catchment area | Marchi and Dalla Fontana (2005) |
| Channels | Vertical distance to channel network | Conrad et al. (2015) |
| Location | Latitude | |
| | Longitude | |

high sensitivity to fluctuations in the training data. The resulting models have a high variance and are unstable to small changes in the data.

RF tackles the shortcomings of RTs by using two expansions. Instead of building a single tree, RF uses the ensemble method bagging which constructs several trees based on bootstrap samples of the data. The resulting averaged prediction has a lower variance and thus increased model stability. Although randomness is added to the procedure through resampling of the data, the underlying predictor-response relationship is not altered by bagging. As a consequence, many of the trees share similar structures. This correlation between trees can lead to a decrease in predictive performance of the ensemble (Breiman, 2001). To introduce diversity to the ensemble and decorrelate the trees, RF is extended by a random feature selection. Instead of using the entire set of predictors to build a tree, a random subset of the predictors is used for each tree. This reduction of predictors leads to a trade-off between the strength of individual trees (high number of predictors) and more diversity between trees (low number of predictors). The respective tuning parameter which controls this trade-off, is mtry, the size of the predictor subset.





Further parameters include 'ntree', the number of trees and 'nodesize', the minimum number of samples to be kept in a terminal node of the trees (Were et al., 2015).

For the interpretation of the RF models, the model function calculates a variable importance measure. This is done by building models which use permutations of a predictor variable. The accuracy of the permuted model is then compared to a
model built from the original data. The returned value indicates the decrease on prediction accuracy after permutation.

### 2.2.2    Cluster analysis

A cluster analysis (CA) was conducted for several purposes:

- CA-1: feature selection

- CA-2: landscape stratification for subsampling

– CA-3: data stratification in CV approach

in CA-1, k-means clustering was used to split the soil texture data of both depth levels into three clusters (Fig. 2). The clustering was performed with the kmeans function using 40 initializations with each 30 iterations. Data were center-scaled. The resulting clusters' predictor ranges at the assigned soil survey sites were retrieved.

CA-2 was applied for landscape stratification on behalf of the gridded continuous multivariate predictor data. The data
of the environmental variables of the study area, however, has certain traits which may hinder or prohibit cluster analysis. The environmental data has high dimensionality with correlating variables, and consists of numerical as well as categorical covariates. These issues were tackled by applying a Factor Analysis of Mixed Data (FAMD) from the FactoMineR package (Lê et al., 2008) on the data set. Because of the high resource demand of conducting an FAMD on the whole data set (33 million gridcells), the function was applied on a random data subset of 100,000 samples (data minimum and maximum additionally
included) first, and then the resulting FAMD model was applied to the whole data set. Additionally, one sample for each class of every categorical variable was also added to the subset. Similarly to a principle component analysis (PCA), the FAMD returns $n$ components, with the percentage of explained variance decreasing with every component. A reliable method to determine the number of dimensions to retain from the FAMD is the 'elbow' approach (Linting et al., 2007). The contribution of each retained dimension to the percentage of explained variance decreases strongly with the first dimensions, until it reaches a
nearly constant value. The 'elbow' approach suggests using all dimensions before the stagnation of the explained variance. The resulting FAMD transformed data was clustered using k-means in CA-2. The number of clusters was determined by the use of cluster validation indices calculated with the NbClust function from the package (Charrad et al., 2014). NbClust calculates 27 clustering indices for each clustering solution in a given range of number of clusters. All of the clustering indices cast their vote for their favoured number of clusters. Because of the high computational cost of NbClust, it was not possible to apply the
function on the whole data set. Instead, repeated random sub-sampling was used to calculate the indices. A random sample of 2,000 data points was drawn from the FAMD data (33,000,000 observations), and a table containing the number of votes for the number of suggested clusters was created. The random sampling was repeated 2,000 times and the votes of each solution added to the table of votes. The number of clusters to be investigated by NbClust ranged between 2 and 17.



CA-3 was conducted in order to perform a stratified CV. The legacy data set including sand, silt, and clay content was
clustered into five equally-sized subgroups to form the strata for model tuning and evaluation. The clustering was achieved by
using a same-size k-means algorithm (Schubert and Zimek, 2019) to divide the profiles of both depth levels into five clusters
based on the soil texture. Finally, the samples of each cluster were randomly assigned to each fold.

### 2.2.3 Feature selection

The RF algorithm is relatively robust against uninformative predictors by only selecting the strongest predictors as splitting
criterion (Hamza and Larocque, 2005; Kuhn and Johnson, 2013). Although the reduction of the predictor set may not necessar-
ily lead to a reduced error, it can still benefit model interpretability and reduce computational time (Chandrashekar and Sahin,
2014).

CA-1 was not only conducted to remove uninformative predictors, but to study the relationship between the environmental
variables and the response value. The clustered profiles were paired with the corresponding predictor values, and the numeric
predictors are tested for normality with the Shapiro-Wilk test. The Kruskal–Wallis test is used to compare the distributions
of predictor values between the three clusters. The resulting p-values were adjusted for multiple comparison by controlling
the false discovery rate (Benjamini and Yekutieli, 2001). All predictor values with significant differences in means ($\alpha = 0.05$)
were used as predictors for the RF models of the respective depth levels. To gain further insight in the predictor-response
relationship, the Dunn's test was performed on the significant response variables as a post hoc analysis.
Preliminary results have shown, that categorical predictors and the usage of the Cartesian coordinate space can lead to
artifacts in the maps of predicted soil texture. Two more models were built in addition to the model using the full predictor set
(full) in order to tackle this problem. One model is leaving out the petrography and genesis layers as predictors (no geo) and
the other is leaving out petrography, genesis, longitude and latitude (no geo+coords).

### 2.2.4 Strategies for unbalanced data

Statistical sampling from the soil data set was used in order to create training and validation data better balanced with regards
to landscape features corresponding to the interaction of the soil forming factors. Please compare Ließ (2015, 2020) concerning
a detailed discussion of this aspect. This was done by applying four subsampling approaches to the model training data based
on the landscape strata obtained from of CA-2. Performance of the models trained on the thereby adapted data was compared
to that of models build with the legacy dataset in its original distribution. Subsampling was conducted to match the spatial
coverage of the landscape strata (area-weighted method = AW), or in order to provide a sample that represents each landscape
stratum with the same amount of data (equal number approach = EN) (similar to Heung et al. 2016). The subsampled dataset
is obtained either by oversampling or undersampling (He and Garcia, 2008). Oversampling obtains the dataset by including all
samples from all strata and then replicating certain randomly selected samples until the desired sample size for each stratum
is reached. Undersampling includes all samples from the minority stratum, and then randomly draws samples from all other
strata, until the desired sample size is obtained. The four applied random stratified sampling approaches are displayed in Fig.
3).





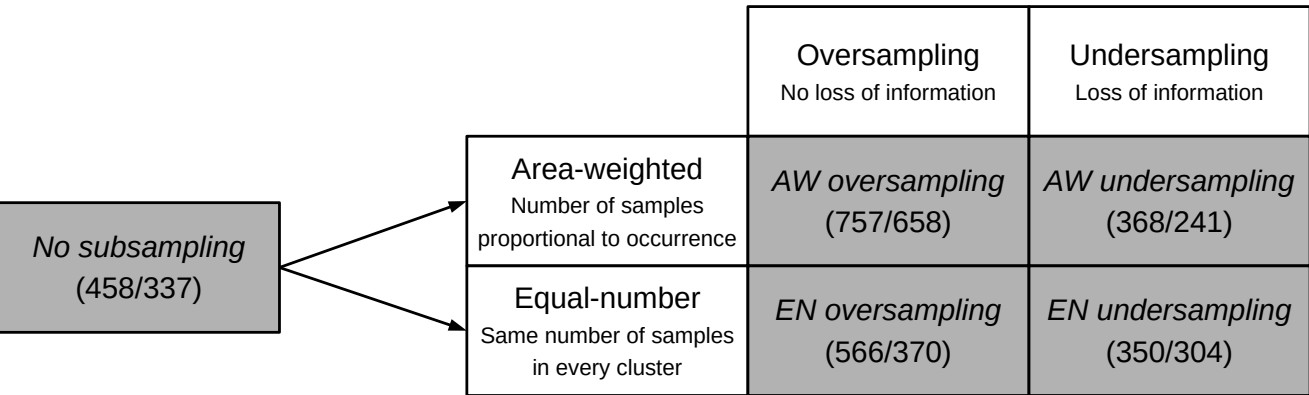

**Figure 3.** Applied sampling approaches. Parentheses show the number of samples in the training set (10 cm and 70 cm depth respectively). Abbreviations of the sampling methods as used in the results are shown in italic.

### 2.2.5 Model tuning and evaluation

Model tuning and evaluation for the RF models was conducted by a nested approach of repeated stratified 5-fold CV (5 repetitions). The detailed procedure is shown in Fig. 4 a. As performance measure, the root mean square error (RMSE) was

derived. In order to make the model performance values comparable for all models, the respective test set was kept the same, while data subsampling was only applied to the respective training sets (Fig. 4 b). Furthermore, response data was centered and scaled (SD = 1) to allow for the comparability of model performance between models targeting sand, silt, and clay content. The k folds of the nested approach were derived by stratified sampling regarding the response data. In order to stratify the dataset regarding all three response variables at once, response strata were formed by applying CA-3. Tuning takes place in the

inner CV, where the model is evaluated for mtry parameter values within the range of 5 to 25, while ntree was set to 1000 and nodesize to 5. Overall, the model building procedure is applied six times in order to create individual RF models for each of the three particle sizes for the two soil depths.

For the data analysis and modeling, the R version 3.5.1 was used (R Core Team, 2018). All computation was performed on a machine running Windows Server 2016 Standard with four Intel Xeon Processor E7-8867 v4 and 6.00 TB of memory.

## 3 Results and discussion

### 3.1 Exploratory data analysis

### 3.1.1 Feature selection

The soil profiles used for model building were split into three groups based on their soil texture with CA-1. A clayey cluster was, thereby distinguished from a silty and a sandy cluster. Primarily, this was done in order to understand which predictor

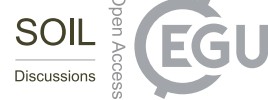

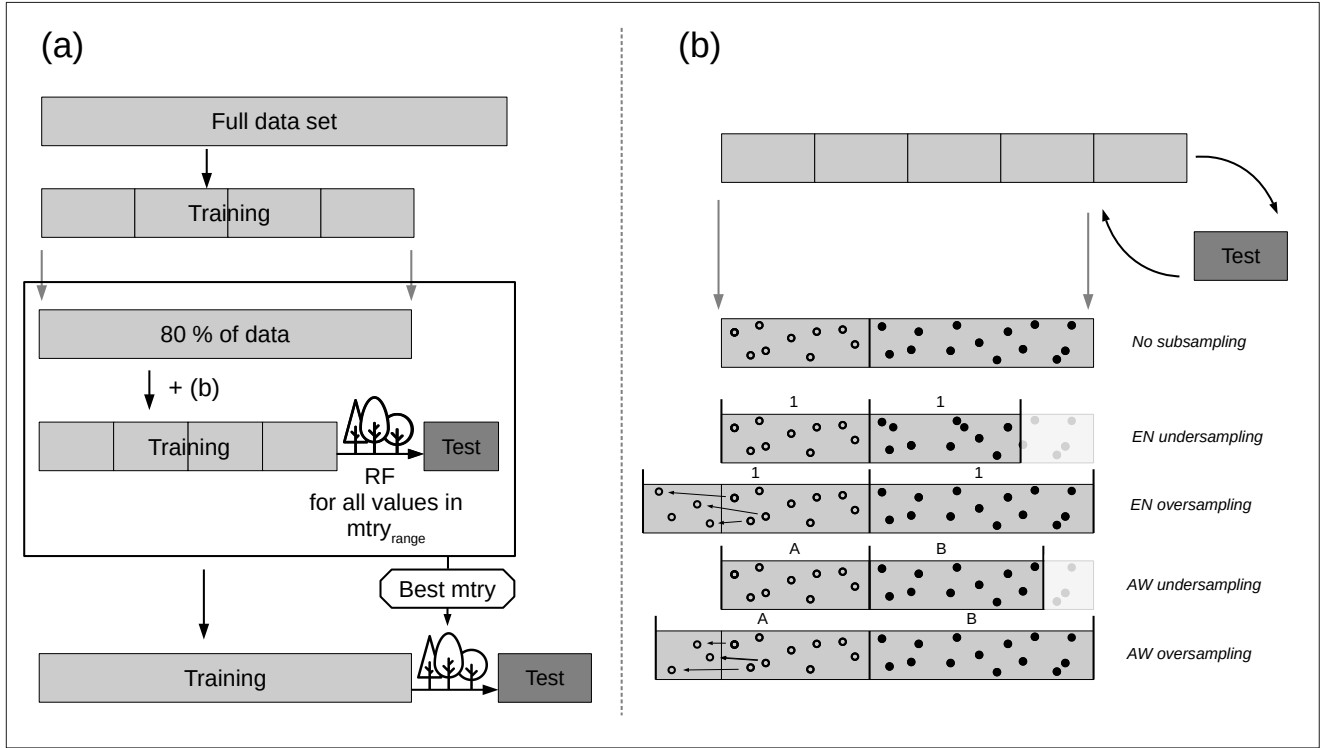

**Figure 4.** Nested k-fold CV approach for model tuning and evaluating. General approach without data subsampling (a). Incorporation of the subsampling strategies in the CV approach (b).

variables are best in separating these three groups and therefore, are expected to have a high explanatory power in the models to predict spatial soil texture distribution within the investigation area. The soil texture of the profiles at 10 and 70 cm depth and their cluster affiliation is shown in Fig. 2 (a) and (b). The spatial distribution of the clusters is shown in Fig. 2 (c) and (d). The distribution of cluster affiliation within the study area shows that most of the profiles in the lowlands belong to the silty cluster. This is typical for the soils of this area which are influenced by loess deposits.

The predictor data at each sampling site was assigned to the soil profile data. The data distribution of the predictor variables between the three soil texture clusters was then compared by applying a Kruskal-Wallis test. Out of 39 numeric predictors, 27 predictors showed a significant difference of the mean at either 10 or 70 cm depth (Table 2). The predictors displaying significant differences between any two of the soil texture clusters were included in the random forest models. A post hoc test was applied to determine which clusters show significant differences concerning the particular predictor variable. The trends in

differences between the clusters are predominantly in agreement across the two depth levels (Fig. 5). For many of the predictor values with significant differences in the means, it was the silty cluster that was the most distinguishable from the other two clusters. From the 54 statistical tests (27 significant predictors for two soil depth), 51 showed differences between the sandy and




the silty clusters and 28 showed differences between the clayey and the silty clusters, while only 19 tests showed differences between the sandy and the clayey clusters.

**Figure 5.** Data distribution of selected predictors per soil texture cluster and soil depth. Letters above bosxplots denote significance groups within one depth level. The Y-axis are cropped to highlight the interquartile range.

Since the clustering application used here for feature selection is a filter method, it is unable to take interactions between different predictors into account. This could compromise the efficacy of the feature selection if there are predictor response relationships which are only revealed in combination with other predictors. The advantage of the clustering method is to create meaningful categories in the data and investigate their relationship with the predictor values, which can not be provided by a wrapper method.





**Table 2.** Predictors included in the random forest model for 10 and 70 cm depth denoted by '*'.

| Predictor | Depth | | Predictor | Depth | |
| --- | --- | --- | --- | --- | --- |
| | 10 cm | 70 cm | | 10 cm | 70 cm |
| Aspect | | | Petrography (GUEK1000) | * | |
| Contributing area | | * | Petrography (GUEK200) | * | * |
| Convergence index (10 m radius) | | | Positive openness | * | * |
| Convergence index (100 m radius) | | * | Sandbin (GUEK200) | * | |
| Convergence index (200 m radius) | | | Siltbin (GUEK200) | * | * |
| Convergence index (500 m radius) | | * | Claybin (GUEK200) | * | |
| Convexity (10 m radius) | | * | Slope | * | * |
| Convexity (100 m radius) | * | * | Slope height | | |
| Convexity (200 m radius) | * | * | Standardized height | * | * |
| Convexity (50 m radius) | | * | Stream power index | | |
| Diffuse insolation | * | * | Terrain classification index | * | * |
| Direct insolation | | | Terrain surface texture (10 m radius) | | |
| Elevation | * | * | Terrain surface texture (100 m radius) | * | * |
| Genesis (GUEK 1000) | * | * | Terrain surface texture (200 m radius) | * | * |
| Land cover 1990 | * | * | Terrain surface texture (50 m radius) | * | |
| Land cover 2000 | * | * | Topographic position index (0-100 m) | | * |
| Land cover 2018 | * | * | Topographic position index (100-200 m) | | * |
| Latitude | * | * | Topographic ruggedness index (10 m radius) | * | * |
| Longitude | * | * | Topographic ruggedness index (100 m radius) | * | * |
| LS-Factor | * | * | Topographic ruggedness index (200 m radius) | * | * |
| Mass Balance Index | | | Topographic ruggedness index (50 m radius) | * | * |
| Mid-slope position | | | Topographic wetness index | * | * |
| Negative openness | * | * | Valley depth | | |
| Normalized height | | | Vertical distance to channel network | | |

### 3.1.2 Landscape stratification for subsampling approaches

CA-2 was conducted in order to subsample from the legacy soil data set and create a balanced model training set. The FAMD data transformation showed an increased drop of explained variance with the sixth factor, resulting in the five first factors being used as input for CA-2. The NbClust application resulted in 29 % of the votes being appointed to finding two clusters in the data, while the second largest vote was 13 % in favor of three clusters. Hence, the environmental data of the study area was stratified into two clusters using k-means. The resulting clusters broadly divided the study area into the mountainous region and the lowlands (compare Fig. 1). This way of stratifying the landscape is an apparent choice since the relatively low number of training samples suggests taking a small number of clusters to have sufficient samples per cluster. Further, the heuristic approach of dividing the landscape, which is often superior to automated classification (MacMillan et al., 2004), also suggests the separation between the high- and the lowlands due to the relatively sharp divide.



## 3.2 Model development

### 3.2.1 Model performance

The predictive performance of the RF models was investigated under different subsampling approaches, a range of mtry tuning parameter values and three predictor sets. Since the RMSE values for model evaluation were calculated for the modeled variables scaled to an SD of one to provide a comparable metric, the RMSE values can be interpreted as zero being perfect predictability, and values over one meaning a worse performance then using the observed mean as the predicted value. The RMSE values of all subsampling approaches for the full predictor set is shown in Fig. 6. The median RMSE is between 0.67 and 0.94, with the silt and sand models clearly outperforming the clay models. For all particle size classes, model performance is better for 10 cm compared to 70 cm depth, with an average difference in the RMSE of 0.08 for clay and silt, and 0.12 for sand. This decrease of performance may be due to a decrease in sample size with soil depth. Studies where sample size has been consistent along profile depth have shown that the predictive performance does not necessarily decrease with soil depth (Adhikari et al., 2013; Vaysse and Lagacherie, 2015).

There is no consistently better performing subsampling method. However, both undersampling approaches seem to have higher median RMSE values than the two oversampling methods. It seems likely that the decline in model performance was due to the reduction of the sample size. Using the RMSE of the whole study area as the selection criteria for the subsampling approach also has its limitations because it does not provide information on the spatial distribution of the prediction accuracy. Adding more weight to samples of a certain cluster can lead to increased accuracy in the respective area while this gain is not necessarily covered by the validation data. The role of subsampling on the distribution of prediction accuracy is exemplarily displayed in Fig. 7. Although there are strong differences of the overall accuracy between clusters, neither of them profit explicitly from a certain subsampling method. The right choice of the subsampling method most likely depends on the underlying data, since other DSM studies have not revealed a distinctly better performing method. While the EN approach increased model accuracy for the minority class in Heung et al. (2014), Schmidt et al. (2008) found the contrary effect in their study and Moran and Bui (2002) found AW to be the best performing model. Sharififar et al. (2019) used a combination of over- and undersampling to create a balanced data set which significantly improved model performance, while over- and undersampling decreased model performance in Taghizadeh-Mehrjardi et al. (2019).

In order to prevent the occurrence of artifacts, predictors have been retained from model building. This led to a decrease in model performance across all particle sizes and depth layers (Fig. 8). The no geo+coords models showed an average increase of scaled RMSE of 3, 7 and 12 % for sand silt and clay at 10 cm depth when compared with the full model.

The $R^2$ values for model performance of the full and the no geo+coords models are shown in Table 3. Model performance of the silt and sand models at 10 cm depth are comparable with the results of Vaysse and Lagacherie (2015) and de Carvalho Junior et al. (2014), while other publications have shown that $R^2$ values above 0.5 are achievable (Moore et al., 1993; Gobin et al., 2001; Adhikari et al., 2013). Moore et al. (1993) argues that $R^2$ values above 0.7 are not to be expected due to the underlying random variability of soil and limitations in the accuracy of measurements. Differences in model performance are most likely



**Figure 6.** Model performance as boxplots of RMSE of the 5 repetitions for three particle size contents for (a) and (d) clay, (b) and (e) silt and (c) and (f) sand and five subsampling methods. The models (a), (b) and (c) are for 10 cm depth while (d), (e) and (f) are 70 cm depth. The subsampling method with the lowest median RMSE is highlighted. The black horizontal lines stand for an RMSE of one, which equals the RMSE of predicting the observed mean.

to be related to the size and the heterogeneity of the study area and the quality of soil samples. This is illustrated well in Fig. 7 which demonstrates the variability of predictive performance across landscape types.

**3.2.2 Model specification**

The mtry values for the full predictor model are shown in Fig. 9. There is no clear trend of optimal mtry value with model performance, and many models have a relatively large range of selected mtry values. It is worthwhile to mention, though, that for certain models the selected mtry value is right at the lower boundary of the tested mtry parameter range, which is the case





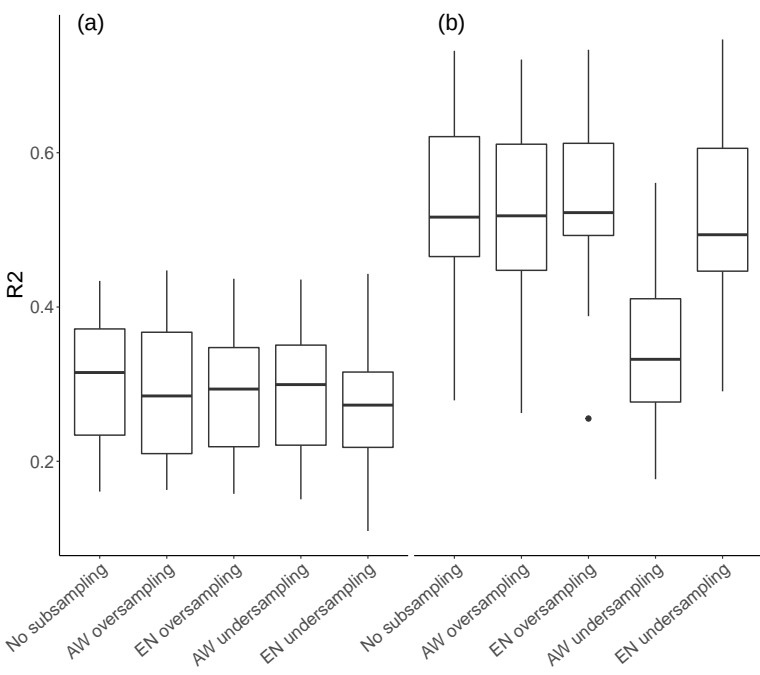

**Figure 7.** Stratum-specific model performance of the silt model at 10 cm depth with (a) showing the $R^2$ of the samples from the lowlands cluster and (b) the samples from the mountainous cluster.

**Table 3.** Model performance in $R^2$ for three texture classes on two depth levels and for two predictor subsets.

| Predictor set | Depth | Particle size | | |
| | | Clay | Silt | Sand |
|---|---|---|---|---|
| full | 10 cm | 0.29 | 0.48 | 0.50 |
| | 70 cm | 0.16 | 0.36 | 0.32 |
| no geo+coords | 10 cm | 0.25 | 0.40 | 0.37 |
| | 70 cm | 0.11 | 0.30 | 0.27 |

for the silt model at 10 cm. Accordingly, an extension of this lower boundary and the corresponding lower model complexity would likely have resulted in even better model performance.

Predictor importance is shown in Fig. 10 a and b. The better model performance at 10 cm depth is reflected in the overall higher importance values. Altogether, petrography has the highest explanatory power. It should be noted, though, that GUEK 200 petrography was included for both depths, GUEK 1000 petrography was only included in the model to predict soil texture at 10 cm depth (Table 2). There are few remaining predictors with notably increased predictive ability. These are latitude for



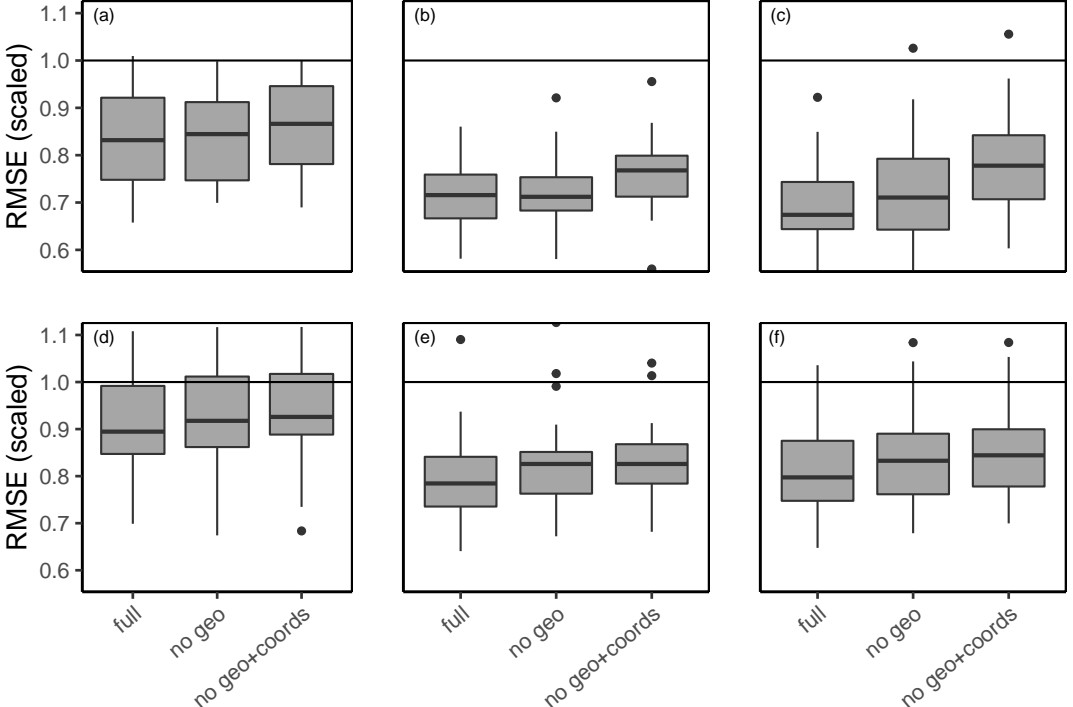

**Figure 8.** Model performance as RMSE for three particle size contents for (a) and (d) clay, (b) and (e) silt and (c) and (f) sand. The models (a), (b) and (c) are for 10 cm depth while (d), (e) and (f) are 70 cm depth. Models were built using either all predictors (full), leaving out the geologic predictors (no geo) or leaving out geology, latitude and longitude (no geo+coords). The black horizontal lines stand for an RMSE of one, which equals the RMSE of predicting the observed mean.

silt and sand, positive openness and the topographic ruggedness index (100 and 200 m radius) for sand and terrain surface texture (200 m radius) for clay.

Omitting the geologic information and coordinates leads to an overall increase of importance values for the remaining predictors (Fig. 10 c). The importance value of elevation increased strongly. The same applies for many other topographical predictors, although in a less pronounced manner (positive and negative openness, diffuse insulation, terrain surface texture

(200 m radius)).

### 3.3  Spatial prediction

Model output was generated by taking the median of all 25 models (CV procedure with 5 folds and 5 repetitions). The predicted, spatially continuous values of the sand, silt and clay content at 10 cm depth corresponding to the models with the best median predictive performance (Fig. 6) are shown in Fig. 11. It needs to be noted that the maps of predicted values are showing the

results of independent models for different soil texture classes, and the results don't add up to 100 %. The method to scale the



**Figure 9.** Results of the model tuning procedure to find the best performing mtry values (mtry$_{select}$) under different subsampling approaches. (a), (b) and (c) show the results of the clay, silt and sand models at 10 cm depth, while (d), (e) and (f) show the same texture classes at 70 cm depth. Grey lines correspond to the tested mtry parameter range

data to 100 % should be selected with the purpose of the specific data utilization in mind. Different approaches could include leaving one of the texture classes out and summing up to 100 %, weighted scaling by texture class or weighted scaling by the regional accuracy of the texture classes.

In the predicted spatial distribution of the sand and silt content, there is a strong regional difference between the lowlands and the mountainous region in the southwest. Sand content is generally increasing with elevation, and is very high in riparian regions and valley bottoms. High silt contents can be expected in the lowlands outside of riparian regions. The spatial variability of all three target variables is dominated by categorical predictor traits (petrography) that draw clear boundaries and even transfer artefacts present in the geological map products. However it is more evident in the sand and clay model output. A limitation of





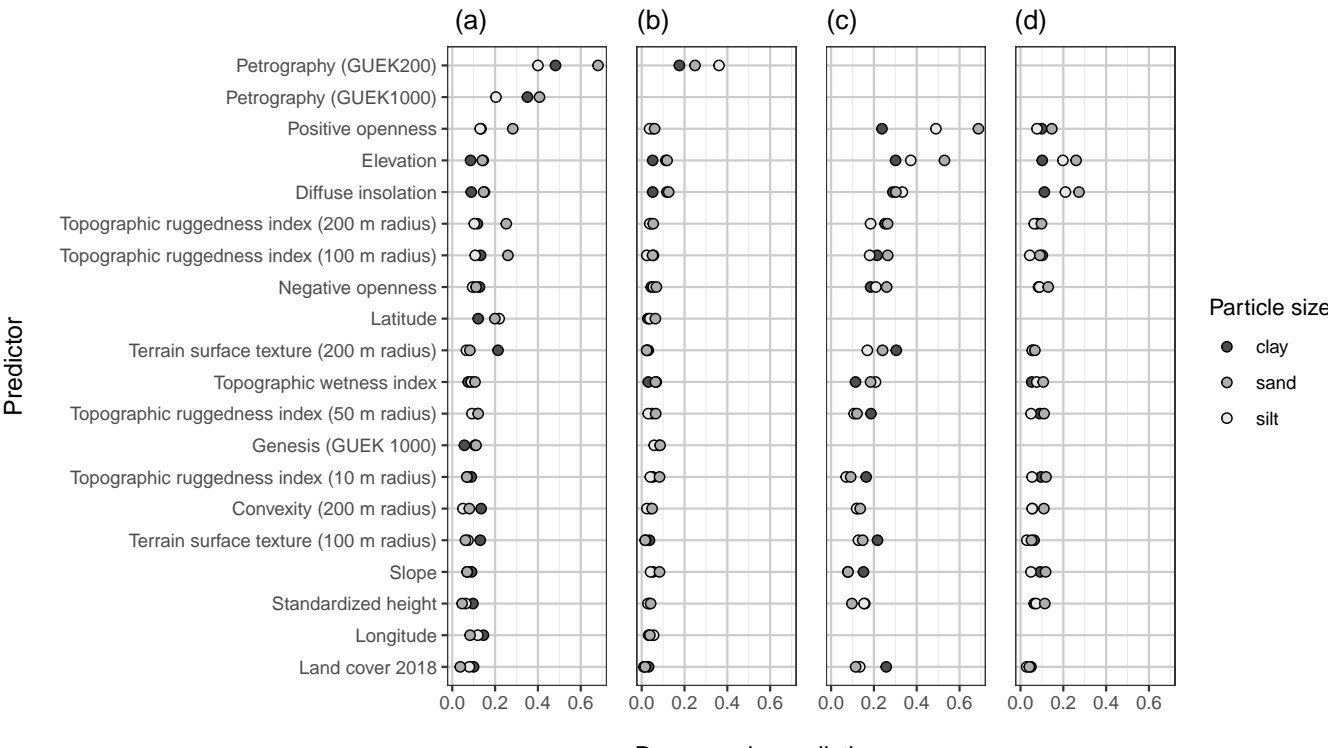

**Figure 10.** Mean importance of the 20 strongest predictors of the model using the full predictor set (a) and (b) and the model leaving out the geologic maps and coordinates as predictors (c) and (d). (a) and (c) show importance values for the models at 10 cm depth and (b) and (d) at 70 cm. Predictors are sorted by decreasing mean importance value. The importance metric is calculated as the decrease in prediction accuracy after the permutation of the predictor values.

the geologic maps, which is the lack of unity in the naming of feature classes between different geographic regions, but also

at federal state boundaries for the GUEK200 is also reproduced in the results. While the GUEK 1000 was generated by the German Federal Institute for Geosciences and Natural Resources (BGR), the GUEK 200 is a joint product between BGR and the regional geological survey institutions. Although the feature shapes align across the tile boundaries, their class description may differ because GUEK200 harmonization at national level is not yet completed. This leads to an abrupt change of predicted sand and silt values in an otherwise homogeneous region (Fig. 11 areal zoom).

These model outputs clearly show the limitation in predictive capacity due to the limitations in the available covariates to represent the parent material. The prediction of soil texture is predominantly based on parent material, which allows to distinguish the observed variability of soil texture between the lowlands and the mountains. Once parent material and coordinates are removed, the models increase the importance of those topographic predictors which can be used to distinguish between these broad geographic regions (elevation, positive openness, diffuse insulation). Pedogenetic processes related to topography

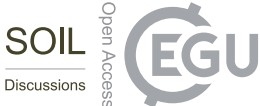

**Figure 11.** Median of all predictions for sand, silt and clay content at 10 cm depth. The maps show the output of the models built with the predictor set specified in Table 2. The scale bar shows distance in meters.

like the lateral redistribution of particles along slopes can only play a minor role, as the low importance values of predictors based on immediate pixel neighborhood have low importance values. However, other DSM approaches have successfully captured relief based variability of soil texture on the scale of hillslopes using only topographical predictors (Moore et al., 1991; De Bruin and Stein, 1998; McBratney et al., 2000).

       The inclusion of expert knowledge such as geological map products in machine learning models for the spatial continuous

soil prediction at high resolution still requires further investigation. While the geologic maps have strong predictive power, they consist of too many different geologic units. This leads to some of the units not having a sufficient number of soil samples to be able to generalize for that unit. However, our approach of reducing the number of geologic units by creating the particle size





class bins was not able to produce useful predictors. One approach could be to use expert knowledge to merge geologic units that have parent material with similar soil texture classes. This merging should happen under the restriction that the resulting

units should be as homogeneous as possible while providing enough samples for training and validation. Solving the issue of abrupt change in predicted values across geologic units could possibly be addressed by a fuzzy approach. Additionally, knowledge on the level of certainty of boundary demarcation between geologic units could be used to create fuzzy geologic maps as predictors.

In order to tackle the issue of artefacts present in the model output, two more models with a reduced set of predictors were

built. The models using all predictors as specified in Table 2 (full) were compared to models leaving out the geologic predictors (no geo) or leaving out geology, latitude and longitude (no geo+coords). Although the dominance of the categorical predictors on the model output was lifted in the 'no geo' model version, an artifact due to the predictors longitude and latitude emerged. This new phenomenon appeared as a horizontal or vertical abrupt change in the predicted values across major parts of the study area (not shown). This aspect has already been observed in other DSM applications employing recursive partitioning

algorithms (Behrens et al., 2018; Hengl et al., 2018; Nussbaum et al., 2018). Møller et al. (2019) addressed this problem with oblique geographic coordinates and provide an overview on ready applied approaches. Accordingly, we tested the usage of three euclidean distance fields instead of Cartesian coordinates. However, the use of this alternative coordinate system led to the emergence of radial artifacts (results not shown).

The additional omission of latitude and longitude from the predictors leads to smoother maps, where only minor abrupt

boundaries exist due to land cover which is also a categorical predictor (Fig. 12). However, this is aspect has to be differentiated from that of the geologic predictors. CORINE land cover classes were classified in remote sensing data products. Hence spatial class boundaries do not reflect expert knowledge. Abrupt changes might in fact be due to land cover changes. The large agricultural fields of the lowlands are heavily impacted by wind erosion of the loess material during bare soil conditions. These 'no geo+coords' predictions are reproducing the spatial variability of the 'full' model even on relatively small scales. Strong

deviations between the two model versions are in the eastern Harz region and in the riparian zones of the southern lowlands.

The difference in sand and silt content between the Harz and the lowlands were most likely derived from the predictors elevation and positive openness. These predictors are strongly correlated (-0.95), have high importance values in both predictor sets and show strong significant differences between the texture clusters (Fig. 5). The other predictors of Fig. 5 have lower values of absolute correlation with elevation (0.27 – 0.37) while still having a significant effects on the texture clusters. These

predictors were more likely related to the variability within the two large-scale regions. The output of the 'no geo+coords' models show much more variability on smaller scales then the 'full' models.

## 4 Conclusions

Our DSM approach has shown that RF is an appropriate method to model the variability of soil texture in the study area. The predictive performance of the silt and sand models is within the range of similar studies, while the prediction of the clay content

did not seem feasible.



**Figure 12.** Median of all predictions for sand, silt and clay content at 10 cm depth. The maps show the output of the models built without using the geologic maps, longitude and latitude as predictors. The scale bar shows distance in meters.

Clustering applications appear to be a versatile tool to be employed at various steps of the DSM procedure. Using a clustering application for feature selection offers additional insight into the predictor response relationship, while clustering to conduct a stratified CV allowed for a robust model evaluation. Overall, stratified k-fold CV is common in DSM. To use the described cluster application allows for a simultaneous stratification regarding multiple target variables. However, to truly evaluate the power of this filter method it would have to be compared with other feature selection methods which would have exceeded the workload for this study. We intend to do so in future studies.

The biggest area of application for data clustering in DSM appears to be in landscape stratification. Dividing the landscape into homogeneous subgroups allows to address the imbalanced learning problem and gives control and feedback over the



spatial distribution of model performance. The resulting stratification of the study area has further potential, like the use of
landscape classes as predictors, the construction of individual models per landscape type or to interpret the predictor response
relationship in different landscapes. A remaining difficulty in clustering applications is the determination of the number of
clusters. Here, the combinations of clustering indices and heuristic methods have proven to be useful tools.

Finally, clustering applications could also provide solutions to the problems encountered during this model building process,
like the replacement of the Cartesian coordinates, the inclusion of expert knowledge, pooling geologic units and blurring the
transitions between geologic units.





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
