# Peer review of "On the benefits of clustering approaches in digital soil mapping: an application example concerning soil texture regionalization"

_SOIL, 2020_

## Author Response (AR1)

This manuscript is about using clustering in a number of applications for digital soil mapping. The first application is feature selection of the mapping model. The second application is for dealing with imbalance training data. The third application is to build folds, to be used in model tuning and model prediction validation (cross-validation).

Thank you. This is capturing the rationale behind the manuscript.

Overall I found that in each of the cases, the rationale for clustering was very unclear. I could not find real justification for these approaches. Besides, the description of what has exactly been done is missing. I read some parts several times, but I stopped at the end of the Methods section because I think this study in fundamentally flawed in its objectives.

We thank the reviewer for his/her critical reading of our manuscript. We agree that certain parts of the methodology section need adaptation to improve understandability. Please find our replies to the specific comments below.

Clustering for the feature selection: it is not clear what the authors did, but from their explanation I dot not see the rationale for doing this. The description made in Section 2.2.3 makes no sense to me. None of these tests are needed. Why clustering the soil texture data? How is feature selection made, for each cluster? The authors claim that feature selection is not needed (Introduction), but then make cluster for feature selection. How has clustering anything to do with feature selection?

We describe the reasoning behind feature selection and the different approaches in the introduction (lines 40 – 45). In this study we use a filter method. Filter methods look at every predictor individually and reject all predictors that don't show an effect on the target variable. Here, this aspect is addressed by finding a difference in predictor values between different groups of soil texture classes. We adapted Sections 2.2.2 and 2.2.3 to improve understandability.

Clustering for imbalance learning problem. Not sure why this is actually a problem. The introduction seems to mix several concepts, such as bias, representativity, predictive performance etc. All papers cited are about categorical mapping, where class imbalance is indeed a problem. I have never heard that imbalance data in a regression setting is a problem. How is imbalance defined in a regression perspective, where classes do not exist? Is a cluster a class? The authors in Section 2.2.4 discuss about strata, not clusters, the two are different.

In digital soil mapping we use machine learning to extract knowledge on spatial soil distribution from data. In fact we build a model reflecting the functional relationship between soil characteristics and the soil forming factors. The model is then applied to extrapolate from the data to the landscape and generate a raster of continuous soil variation. However, this is only justified, if the dataset is representative of the landscape's spatial variability. This is why we use statistical sampling approaches to collect soil data in the first place. But, ready-available legacy soil data often does not comply with this request. Still, we may use statistical sampling approaches for resampling from the data to improve the areal representativeness as suggested in Ließ (2015). We agree that the use of the term "imbalanced learning" is misleading in this context and adapted the manuscript accordingly.

Clustering for building robust models. I would argue that the approach used by the authors is not correct. I have seen papers dealing with building geographical strata to define the CV folds, but the authors used the response variable for this. It does not make sense to me. The risk with building clusters on the training data is that large

part of the feature space covered by the data within the cluster will be missing to the model when the model is fitted, so that there is a serious risk of extrapolation when predicting in that fold. Random k-fold is preferred for this, because each fold with contain values that are likely to be different.

The reviewer is referring to our cross-validation approach described in lines 179 – 182. Stratified cross-validation on behalf of the response variable is common practice in machine learning. In this particular case we use a cluster analysis to generate the strata of the response variable which are then randomly sampled to generate the folds for cross-validation. In detail: We created 5 equally sized clusters based on the target variables. However, we did not use these clusters as folds, but used them for a stratified sampling approach to create balanced folds with an equal number of samples from every cluster in each fold. We adapted this text section to improve its understandability (track-changes doc, lines 193-196)

In addition, the authors are constantly mixing the terms between clusters and strata. They are different terms with different meaning.

In fact, we use the cluster analysis for stratification: We generate strata for stratified cross-validation (CA-3) and we generate landscape strata for stratified random sampling (CA-2). We will adapt the respective text sections to improve understandability.

This manuscript is about clustering, but the clustering is poorly described and often not done in the correct way. For example, the correlation among variables is not accounted for.

Correlation is only an issue for CA-2, but not for CA-1 and CA-3. For CA-2, the correlation between predictors was taken into account by transforming the data with the FAMD (updated document lines 159 – 178).

Another example is the number of clusters is determined from a sample of only 2,000 points. Even if it is repeated, this is simply not sufficient for determining the optimal number of clusters.

We have also tested the use of larger samples to determine the number of clusters. However, this resulted in the same number of clusters. In fact, our analysis has shown that a sample size of 2,000 was sufficient to produce a stable outcome when the approach was repeated. We added this information to the respective text section. (track changes lines 190)

Other comments on Introduction and Methods.

Line 32: Do the authors mean linear regression? The other models cited after are also models for regression, (e.g. RF).

Yes, the reviewer is right. We changed regression model to "multiple linear regression".

Line 37: citation to Blanco et al. (2018) does not really support the claim of this sentence, look at review papers.

We agree and removed all single-study references.

Line 37: citation to Møller et al. (2019) does not really support the claim of this sentence, look at review papers.

We replaced Møller et al. (2019).

Lines 39-40: this is a very wrong statement. RF is of course prone to overfitting, like all regression models. It has been well described in the literature. For a recent example, see Makungwe et al. (2021): https://doi.org/10.1016/j.geoderma.2021.115079, but there are many more.

We agree that the original statement is controversial. We have adapted the text accordingly.

Further, how does overfitting relate to the presence of outliers or "parameters". What do you mean by parameters?

We have adapted the text section to improve understandability.

Lines 41-45: I do not understand this part. Feature selection do not reduce the noise cause by uninformative predictors. Feature selection would also not omit predictors of importance because these would damage the prediction accuracy, which machine learning models are concerned with.

The reviewer has pointed out a problematic statement in our manuscript, and we removed that section.

Lines 51-52: Already mentioned at L. 48-49.

We removed the repetition.

Lines 51-53: So in one sentence the problem is the bias and in the next one the problem is the prediction performance of the model. I think the problem is that with oversampling (clustered) data, we might be overoptimistic about the predictive performance of the model. It has nothing to do with getting biased estimates of the property.

The aspects the reviewer is referring are embedded in the following text section (lines 49 – 60):

*"Because soil forming factors operate on different scales, it is important that the spatial distribution of the data is suitable to capture the large- and small-scale variation of soil. Moreover, a bias could be added to the prediction if samples from certain parts of the landscape are over- or under-represented in the data. This would lead to an imbalanced learning problem and compromise the predictive performance of the models (He and Garcia, 2008). In order to construct a model that can effectively predict throughout the landscape, it is important to have a statistically representative sample of training and validation data that allows for the generalisation from the data to the spatial landscape context (Ließ, 2020). The most common approaches in dealing with this issue involve (a) creating a more balanced training set by sampling from the entirety of observations, and (b) cost sensitive learning frameworks, in which the learning algorithm penalizes the prediction error of underrepresented samples (He and Garcia, 2008). Many DSM applications tackle the problem of data imbalance with the subsampling approach (Moran and Bui, 2002; Subburayalu and Slater, 2013; Heung et al., 2016; Sharififar et al., 2019). This can be achieved by stratifying the study area into homogeneous sections, and drawing a certain number of samples from each of these strata."*

As explained in our reply above, we agree that the reference to "imbalanced learning" is misleading in this context and removed the corresponding sentence. We are aware of the possibility to overfit a model by extensive oversampling: On the one hand, we apply a nested cross-validation approach to avoid overfitting on model training and tuning. On the other hand, the oversampling and other sampling approaches (Figure 3) were only applied to the respective training sets as indicated in Figure 4 and described in lines 214-216. This means it does not affect model evaluation and we are not overoptimistic of the predictive model performance.

Lines 53-55: very unclear what the link between the two sentences is.

We are rewriting the passage to allow for a better link of the arguments.

Lines 58-59: to this point I still have no good justification to why the problem of having an imbalanced data is a problem for regression. All the cited papers at L. 58-59 are for classification!

We use resampling to improve areal representativeness as specified above. As resampling is often used to address the imbalanced learning problem in classification tasks we cite the respective literature. However, we agree that we need to emphasize the difference between the two purposes. We, therefore, refrain from using the term "imbalanced learning problem" for our approach and adapted the manuscript accordingly.

Lines 61-63: What does it mean model robustness? The cross validation is only here to estimate error needed to estimate the validation statistics, but the "model robustness" is not evaluated. What does this sentence mean: "the outcome can still be compromised by an uneven distribution of classes between the data subsets", I do not think this paper is about classes?

We applied a nested cross-validation approach. Cross-validation was not only used for model evaluation (outer cross-validation cycle) but also for model tuning (inner cross-validation cycle). In using cross-validation for model tuning we generate robust models. We have rewritten the text section to improve understandability.

Line 65: the authors should systematically differentiate between dependent and independent values. Do they do the clustering on the independent or independent variables? It makes a lot of difference, but both options are possible.

In this study we have performed three different clustering applications, two on dependent and one on independent variables. The details of the individual clustering applications are described in section 2.2.2.

Lines 94-96: Removing a soil sample from the modelling because the sand content is too high is not a good reason. This is clearly not an outlier.

The sample we are referring to is located in a small scale geological anomaly and is surrounded by otherwise silty soils. The correct prediction of the sample could only be based on the geologic unit, however it is the only sample in that class of geologic unit. The modeling set up as we use it could not possibly predict the soil texture at that location because (a) we need >2 samples for training and testing, and (b) the minimum node size of our random forest model is set to 5. This means that inferring soil texture from this specific geologic unit could only take place if more than 5 samples were available. However, keeping the sample in the model building process would come at a high cost, because the model couldn't make the necessary distinction to neighboring samples and would therefore strongly deteriorate the model performance in the proximity of this sample. We do agree though that the term outlier is misleading in this context and rewrote the section.

Lines 97-99: a clustering was done on what? On the predictors or the values of the soil properties? It should be systematically written.

We recognize that the description of the clustering approach was difficult to follow and clarified this in the methods section.

Lines 130-131: this is not an accurate description of what random forest is. Also, RF is based on RT, but the authors do not mention the differences.

The description of RF is in lines 130 – 150. The difference between RF and RT is 137 – 144. However, we adapted the text section to clarify the differences between the two algorithms.

Line 134: rectangular region of the predictor space?

Although the phrase is frequently used in the literature, we see that it is easy to misunderstand here. Instead we changed the text to use the term "abrupt changes with predictor thresholds".

Line 135: I would argue this is not correct. Of course RT are sensitive to the training data, because it is a data-driven model. The text should really be made more precise.

We agree with the reviewer that our description was not clear. We rewrote the section to leave out the shortcoming of RT and emphasize the difference between RF and RT instead.

Line 138: bootstrap sample of the data: training data or predictors?

A more detailed description of the bagging procedure was added to the text to clarify it.

Line 142: the authors should be consistent in the writing: feature, but at other places I see variable, covariate, predictor, predictor variable, predictor data.

We thank the reviewer for this remark and recognize the possible confusion in using multiple terms. The whole manuscript was reviewed in this regard to minimize the number of different terms.

Lines 159-160: what are these traits?

The difficulties in the data structure are described in the section following line 161. We recognize that the transition was difficult to follow and adapted the text accordingly.

Lines 161-166: applying clustering on a mix of categorical and continuous variables is a very complex problem. More information on what has exactly been done is necessary. How are these categorical variables transformed? How can a PCA be made on categorical variables? Also, you should use systematic random sampling, not simple random sampling for collecting a subset. No need to include the minimum and maximum value, and no need to add one sample per class. If the sample is sufficiently large, all classes should be included.

We did not apply a PCA but a Factor Analysis of Mixed Data as described in lines 162 – 178. Simple random sampling is appropriate if the sample size is sufficiently large. On the contrary, systematic sampling usually compromises on the randomness. We have additionally included the minimum and maximum values of all numerical predictors and all classes of the categorical predictors in order to be able to apply the function for FAMD transformation trained on the data subset to the complete dataset. We added this information to the respective text section.

Lines 160-170: Is correlation included in the clustering? The correlation should be accounted for in the clustering process. This can be done by rescaling the predictors by the inverse of Cholesky transformation of the variance-covariance matrix.

The correlation between predictors was addressed by transforming the data with the FAMD (lines 159 – 178).

Line 174: so the number of cluster is actually decided based on 2,000, selected randomly out of the 33 millions of points?

We have also tested the use of larger samples to determine the number of clusters. This resulted in the same number of clusters. Finally, our analysis has shown that a sample size of 2,000 produced a stable outcome when the approach was repeated. We added this information to the text.

Section 2.2.3. None of these normality tests are needed, what is the rationale for doing this.

The reviewer is right, the sentence referring to the normality test was a left-over from a preliminary manuscript version (Lines 189-190). We removed this sentence.

Line 218: the response data?? Why not the predictor variables??

The reviewer is referring to our stratified cross-validation approach. We have answered the question why we stratify on behalf of the response variable previously. Please refer to the respective reply.

**RC2**

We thank the reviewer for his/her attentive reading of the manuscript and constructive suggestions.

Abstract: please add the brief obtained results, e.g., accuracy
We added a summary of the model accuracy to the abstract.

Line 16: add a relevant cit.
We included references to the importance of soil texture maps and the role of soil texture for physical processes.

Line 96: The soil texture of the soil legacy dataset used as model input or model output?
The text was clarified in this regard.

Line 107: please add a relevant cit. and software
The used software are cited in line 116 and Table 1.

Line 156: did you assign the number of clusters into 3?
The description of the clustering application was adapted to clarify this aspect.

Lines 160 to 184: are not clear. Adding a flowchart might enhance the clarification
We considered the option to add an additional figure, but rejected this because of the high number of figures in the manuscript. Instead we clarified the text to make it more precise and allow the reader an easier overview of the FAMD procedure.

Feature selection is also unclear. Did you specify the significant relationships between soil fractions and covariates? Please justify the paragraph
We use a novel approach of a filter method to study the relationship between the individual predictors and the target variables. The corresponding text section was adapted to improve understandability.

Did you transform soil fractions? If you use the raw data, the predicted maps do not guarantee the sum of 100%.
We did not scale the results to 100% because it was in the objective of our study to investigate the individual predictability of each texture class and not to create a ready-to-use soil texture map. There are different ways to scale the data to 100% (scaling all texture classes equally, or scaling the texture classes which have the lowest model performance), and the scaling could be applied to the data to tend to the needs of the individual user. The reasoning for not scaling the results are given in the lines 314 – 318.

Random oversampling is just "copy and paste" of the original data. Do not think if the approach resulted in overfitting?
On the one hand, we applied a nested cross-validation approach to avoid overfitting on model training and tuning. On the other hand, the oversampling and other sampling approaches (Figure 3) were only applied to the respective training sets as indicated in Figure 4 and described in lines 214-216. This means it does not affect model evaluation and we are not overoptimistic of the predictive model performance.

Line 245: please add a relevant explanation about the filter method in feature selection in the method section of the paper
We extended the methods section accordingly.

Please calculate line concordance instead of R squared.

We are using $R^2$ because it is a frequently used metric in model evaluation and therefore allows the comparison of our results with other studies.

I also recommend reducing the size of the paper. There are many aspects of modeling are included in the current version. One possibility is to move them to the appendix.

We are aware that the methods section is rather long. We have thoroughly considered to reduce it. However, the extent of the methods section is due to the methodological focus of the paper. The methods are at the center of its rationale, and all described aspects are important for understandability. Therefore, we refrain from shifting any of them to the appendix.

---

## Author Response (AR2)

The authors did a great job to improve the paper. However, there are still some needed modifications. This should be done for the following questions. I recommend that the authors answer the questions not only in the body of the manuscript but also in the reply to reviewer files, not just mentioning "e.g., Line 316". Added to this, I just looked at my questions:

We thank the reviewers for their analysis of the manuscript and the previous reviews.

**1. The clustering approach is not clear. For example, clustering data into three groups, how does this approach help in modeling?**

The reviewer refers to CA-1. We adapted the corresponding text section in lines 151-154 to add further information with this regards: "in CA-1, k-means clustering was used to split the soil texture data of both depth levels into three clusters to distinguish between sandy, silty and clayey soils (Fig. 2). The distribution of every predictor value among the three clusters is analysed, to a) determine whether the predictor has any influence on soil texture (feature selection in chapter 2.2.3), and b) gain process understanding by analysing the relationship between predictors and soil texture."

**2. Feature selection is also unclear. Did you specify the significant relationships between soil fractions and covariates? If yes, this is not a Filter approach?!**

According to our understanding, filter methods are a broad term for procedures which quantify the individual predictor-response relationship, and define a threshold to "filter"-out weak predictors. One of the more popular filter methods is ANOVA. Kruskal–Wallis is a nonparametric equivalent to ANOVA.

**3. Did you transform soil fractions? If you use the raw data, the predicted maps do not guarantee the sum of 100%. You should transform data and then do modeling.**

The respective contributions of the particle size fractions of the legacy soil data sum up to 100%. Still, training models from these data may result in predictions which do not sum up to 100%. We have addressed this issue in lines 311-315: "It needs to be noted that the maps of predicted values are showing the results of independent models for different soil texture classes, and the results don't add up to 100%. The method to scale the data to 100% should be selected with the purpose of the specific data utilization in mind. Different approaches could include leaving one of the texture classes out and summing up to 100%, weighted scaling by texture class or weighted scaling by the regional accuracy of the texture classes."

**4. Random oversampling is just "copy and paste" of the original data. Do not think the approach resulted in overfitting? I mean that Random oversampling is not a good approach to balance the datasets. Please use another approach.**

Oversampling can be used as a tool to shift the model along the variance-bias gradient: If it is assumed that the machine learning model has a better representation of the majority class, the overall model error can be increased while reducing the error in the minority samples. Please compare Taghizadeh-Mehrjardi et al. (2019), and Sharififar et al. (2019).

**5. Please calculate line concordance instead of R squared. I mean R squared is not a good indicator to show the accuracy of the models.**

We decided for the use of two error metrics: a scaled RMSE and R squared. RMSE allows an estimation of the deviation between observed and modelled results, however R squared is a more

popular metric in the field, that allowed us to compare the model performance with other publications and put our results into perspective.